# The Differential Entropy Generation Rate as a Unified Measure for Both the Stability and Efficiency of an Axial Compressor

**Jingyuan Ma and Feng Lin ***

Marine Engineering College, Jimei University, Xiamen 361000, China; pauline870@126.com
* Correspondence: fenglinfenglin2021@126.com

**Abstract:** Stability and efficiency are the two most important performance indicators of highly loaded aviation axial compressors; however, they often cannot be achieved simultaneously. As an effective means of stability expansion, casing treatment usually damages the peak efficiency. In this study, the differential entropy generation rate was used as a unified measure of stall margin and efficiency, so that both stability enhancement and efficiency improvement could be considered at the initial casing treatment design stage. NASA Rotor 67 was selected, and two single circumferential grooves at two different axial positions were applied, which served as a test case to check how entropy generation rates in the flow field vary with changes in peak efficiency and stall margin. The distribution of entropy generation and differential entropy generation rate were compared with that of the solid casing. The correlation between differential entropy generation rate and the peak efficiency was analyzed, and how the flow mechanism of casing treatments affects entropy generation was determined. Methods for measuring and comparing the impact of different casing treatments on peak efficiency are proposed. At the same time, the distributions of differential entropy generation rate in the near stall were explored, and the relationships between the differential entropy generation rate and the flow structures are detailed. A comparison of the influence of different casing treatment on stability is given with respect to the contours of the differential entropy generation rates. It is demonstrated that the differential entropy generation rate is a unified measure to balance the tradeoffs between the stability and the peak efficiency for different casing treatments for the same compressor.

**Keywords:** differential entropy generation; rotating stall; stability; efficiency; axial compressor





## 1. Introduction

In the field of compressor stability enhancement research, it is often a challenging task to simultaneously achieve stability enhancement and maintain peak efficiency [1–6]. Stability enhancements have been studied for decades; however, there are still challenges that remain unsolved. For instance, casing treatment (CT) is an effective technology and can yield good stability enhancement effects, but most CTs result in a significant decrease in peak efficiency. As a matter of fact, the design focus of CTs is primarily on stability enhancement, which means that during the initial design stage, designers are unable to give efficiency much consideration.

The mechanism of casing treatment for stability enhancement involves modifying the interface between the tip leakage flow and the main flow that influences stall, so as to delay the onset of stall [7]. Stall margin, the margin between the peak efficiency point and the stall boundary, can be evaluated by calculating the entire characteristic of the compressor with the casing treatment, up to the near-stall operating point. To reduce workload and obtain an early prediction of the stability enlargement capability of a designed casing treatment, various physical parameters and methods have been proposed in previous studies [8–12]. Among them, the bell curve method [8] is an effective approach. The bell curve method involves cumulating the axial momentum in the region near the blade tip to obtain the distribution curve along the streamwise chord length, named after its bell-shaped

appearance. The axial position of the peak of the bell curve corresponds to the location where the main flow and leakage flow at the blade tip region intersect. This method is used to predict the stability enlargement effects of different casing treatments, enabling comparison of the effectiveness of different designs. However, this method only measures the stability enhancement capability and does not take into account the impact of the casing treatment on peak efficiency.

When CTs in a compressor lead to significant efficiency loss, remedies must be implemented to reduce this loss, either fine-tuning the geometry of existing CTs or combining them with other techniques. Fan et al. [13] conducted optimization studies on groove-type casing treatments used in transonic rotors. The results showed that by adjusting the structural parameters of the casing treatment, it is possible to further enlarge the stall margin while reducing efficiency losses. However, this optimization approach often requires extensive exploratory calculations and design iterations. Wang et al. [14] proposed a slotted casing treatment coupled with blade tip injection. They utilized tip air injection to enhance the adaptability of the casing treatment and make changes in the compressor flow field, simultaneously extending stability and reducing its negative impact on compressor efficiency. This approach also requires extensive trials to obtain the best peak efficiency while maintaining the stall margin with CTs.

At this point, we can see that there is an ironic cycle in the compressor design and optimization process. When designing a compressor from scratch, one must focus on the peak efficiency first and then optimize the design for the stall margin later. If the stall margin does not meet the given design specifications, CTs should be applied. Then, when designing CTs, the designing sequence is reversed. Stability is the first concern. Once stability is satisfied, one must check the damage to the peak efficiency and optimize the CTs. In this study, we aimed to provide a unified measure that can simultaneously evaluate the efficiency (or the losses) and the stability enhancement capability, so that the designers can reduce time-consuming trials by breaking the ironic cycle.

Entropy is the state parameter of the spontaneous process in accordance with the second law of thermodynamics. Entropy contours are a popular way to observe losses in flow fields. Entropy itself is a thermodynamic state parameter; therefore, it does not directly reflect local losses, which leads to the inability of entropy contours to distinguish between different flow structure losses. As a matter of fact, entropy generation reflects irreversible losses. While the overall entropy generation can be obtained by multiplying the difference between entrance and exit entropy by the mass flow rate, for the in-depth study of internal flow field losses in compressors, a method is required that can directly observe local losses and perform the localized quantification of entropy generation. Entropy generation is a common measure of irreversibility in fluid devices. The differential entropy generation rate (DEGR) method can fit the needs. This method was proposed by Herwig et al. [15–19], who used it for calculating local losses in pipelines and heat transfer. In recent years, Du et al. [20,21] further developed this theory and applied it to the calculation of internal flow field losses in compressors. The DEGR method can provide a detailed observation of the high local losses caused by the casing treatment, thereby optimizing the peak efficiency of the casing treatment. Zhang et al. [21] used the DEGR method and found that some casing treatments with good stability effects can also reduce entropy generation. This suggests the possibility of establishing a connection between stability and entropy generation, which is enlightening for this study. Ma et al. [22] pointed out the dependence of the DEGR method on grids and turbulence models, and conducted a detailed study on the correspondence between different flow structures and the distribution of DEGR, laying the foundation for this study.

It is natural to know that the DEGR can be an excellent measure for localized losses, but evidence is needed to show that it can also be used as a measure for stability. Early research [23–27] in stalling mechanisms concluded that the axial location of interface between the incoming main flow and the reversed tip leakage flow could be used to characterized the onset of stall. Thus, the closer this interface is to the blade's leading

edge, the closer the compressor is to instability. In [12], this interface was identified by calculating the axial momentum in the tip region. Can the DEGR do the same thing? If yes, then the DEGR could be used as a measure for stall margin enlargement, and therefore, the measures for both the flow losses and the stability can be unified. This is the research motivation of this paper.

The research plan of this study starts with a compressor rotor. NASA Rotor 67 (R67), which is well known in the field of compressor research, was chosen in this study. Two circumferential grooves at different axial positions were applied to enlarge the stall margin; their DEGR distributions were compared with those of the solid wall (SW) casing for research purposes. The goal was to explore the relationship between the DEGR and efficiency, as well as the relationship between the DEGR and stability, and to find a way to unify the measure of efficiency and stability using the DEGR.

As a preparation of the research, it is important to ensure the reliability of the grids and the accuracy of the CFD simulations. This involves establishing a compressor grid construction method and an independence verification method applicable to DEGR analysis. This step is necessary because conventional grid independence verification methods are not sufficient for calculating the DEGR. In this study, different numbers of nodes were assigned to different sections of each grid, and grid independence was verified for each region to obtain appropriate grids.

Notably, the work presented here is a feasibility study for the use of the DEGR as a unified measure for both efficiency and stability. For efficiency, although it is natural that entropy generation represents the flow losses, it is necessary to demonstrate this point from both global and localized views. Globally, it is demonstrated that the total entropy generation, an integration of the entropy generation rate on each grid over the computation domain that encloses the compressor rotor, follows a reverse trend to the efficiency when the incoming mass flow varies. Locally, the DEGR contours are to be associated with each flow structure in the flow field. For stability, the fact is that the DEGR is very small in the incoming main flow, yet very large in the region dominated by reverse tip leakage flow. An interface exists across which the DEGR increases rapidly. This is due to the fact that the complex flows, such as shear layers, vortex, shock wave boundary interaction, etc., all happen immediately after the interface. In this study, it was demonstrated that this interface is identical to the interface that characterizes the onset of stall, and thus, the DEGR can also be a measure for stall onset.

This paper is organized as follows: after the introduction, grid independence is extensively verified for the DEGR in Section 2, followed by the numerical scheme verification by comparing the simulation results with the experimental values; in Section 3, the correlation mechanisms between entropy production and efficiency, as well as the correlation mechanisms between the DEGR and the stability, are investigated; finally, Section 4 summarizes the results and concludes that the DEGR is a suitable unified measure of a compressor's efficiency and stability.

## 2. Grid Independence Verification and Numerical Simulation Method Validation

### 2.1. Model and Grid Independence Verification

The classical NASA Rotor 67 compressor was used as the model, as shown in Figure 1, which is divided into three sections: inlet, rotor, and outlet. The inlet and outlet sections were given sufficient length to ensure numerical stability. ANSYS ICEM was used for structural meshing of the compressor, ANSYS CFX was used for numerical simulation, and ANSYS CFX-Post was used for data analysis. The boundary conditions for the inlet and the outlet were set as follows: inlet pressure of 0.101 MPa, total temperature of 288.15 K, and the outlet pressure was varied to obtain different compressor operating conditions. The SST turbulence model was chosen for the steady-state calculation of the single-blade passage of the compressor.

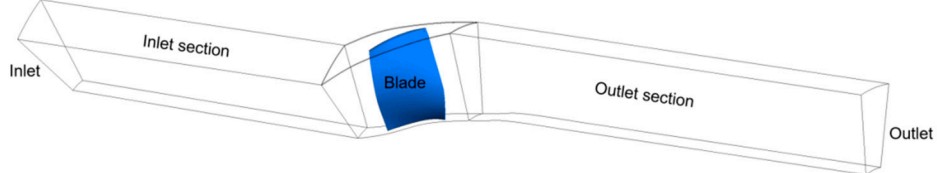

**Figure 1.** Single passage of the R67 model.

The formulas involved in the DEGR used in this study were programmed in ANSYS CFX-Post. The DEGR includes four parts—the equations of which are presented in (1)–(4)—which aligns with the formulation provided in reference [22]. It was shown in [22] that conventional standard grid independence verification cannot meet the requirements of the DEGR. A more detailed grid independence verification is needed, which was refined to verify the independence for each segment of the mesh. Each mesh segment and its labels are shown in Figure 2. The labels of the mesh segment in Table 1 correspond to Figure 2. Three different numbers of nodes are set on each mesh segment (Table 1). After numerical simulation, the total entropy generation in the rotor region was calculated (Table 1), and the distributions of the DEGR along the streamwise direction were determined for different numbers of nodes (Figure 3). The DEGR was calculated by averaging values over the entire span and circumferential direction. The streamwise range of 1.0–2.0 pertains specifically to the rotor domain, which aligns with subsequent research findings from the graph, the influence of adjusting the nodes on each mesh segment can be accurately determined for entropy generation and DEGR. Finally, a grid (a = 177, b = e = 33, c = d = 33, f = g = 33, h = 177) that satisfied the grid independence of DEGR calculation was obtained, with a total of 6.26 million grids. Far fewer grids are required to reach grid independence for conventional parameters such as efficiency, flow rate, and pressure ratio.

$$\dot{S}_{irr,D}''' = \frac{\mu}{T}\left(2\left[\left(\frac{\partial u}{\partial x}\right)^2 + \left(\frac{\partial v}{\partial y}\right)^2 + \left(\frac{\partial w}{\partial z}\right)^2\right] + \left(\frac{\partial u}{\partial y} + \frac{\partial v}{\partial x}\right)^2 + \left(\frac{\partial u}{\partial z} + \frac{\partial w}{\partial x}\right)^2 + \left(\frac{\partial v}{\partial z} + \frac{\partial w}{\partial y}\right)^2\right), \tag{1}$$

$$\dot{S}_{irr,D'}''' = \frac{\mu}{T}\left(2\left[\left(\frac{\partial u'}{\partial x}\right)^2 + \left(\frac{\partial v'}{\partial y}\right)^2 + \left(\frac{\partial w'}{\partial z}\right)^2\right] + \left(\frac{\partial u'}{\partial y} + \frac{\partial v'}{\partial x}\right)^2 + \left(\frac{\partial u'}{\partial z} + \frac{\partial w'}{\partial x}\right)^2 + \left(\frac{\partial v'}{\partial z} + \frac{\partial w'}{\partial y}\right)^2\right), \tag{2}$$

$$\dot{S}_{PRO,C} = \left(\frac{\Phi_\theta}{T^2}\right) = \frac{\lambda}{T^2}\left[\left(\frac{\partial T}{\partial x}\right)^2 + \left(\frac{\partial T}{\partial y}\right)^2 + \left(\frac{\partial T}{\partial z}\right)^2\right], \tag{3}$$

$$\dot{S}_{PRO,C'} = \frac{\alpha_t}{\alpha}\frac{\lambda}{T^2}\left[\left(\frac{\partial T}{\partial x}\right)^2 + \left(\frac{\partial T}{\partial y}\right)^2 + \left(\frac{\partial T}{\partial z}\right)^2\right], \tag{4}$$

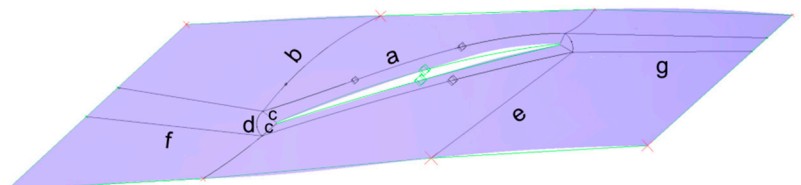

**Figure 2.** The labels of grid divisions.

**Table 1.** Entropy generation corresponding to different grids.

| a | Entropy Generation (W/K) | b and e | Entropy Generation (W/K) | c and d | Entropy Generation (W/K) | f and g | Entropy Generation (W/K) | h (Height) | Entropy Generation (W/K) |
|---|---|---|---|---|---|---|---|---|---|
| 129 | 17.54 | 17 | 16.21 | 17 | 17.07 | 17 | 17.25 | 113 | 16.21 |
| 177 | 17.79 | 33 | 17.79 | 33 | 17.79 | 33 | 17.79 | 177 | 17.79 |
| 225 | 17.78 | 49 | 17.85 | 49 | 17.81 | 49 | 17.67 | 225 | 17.85 |

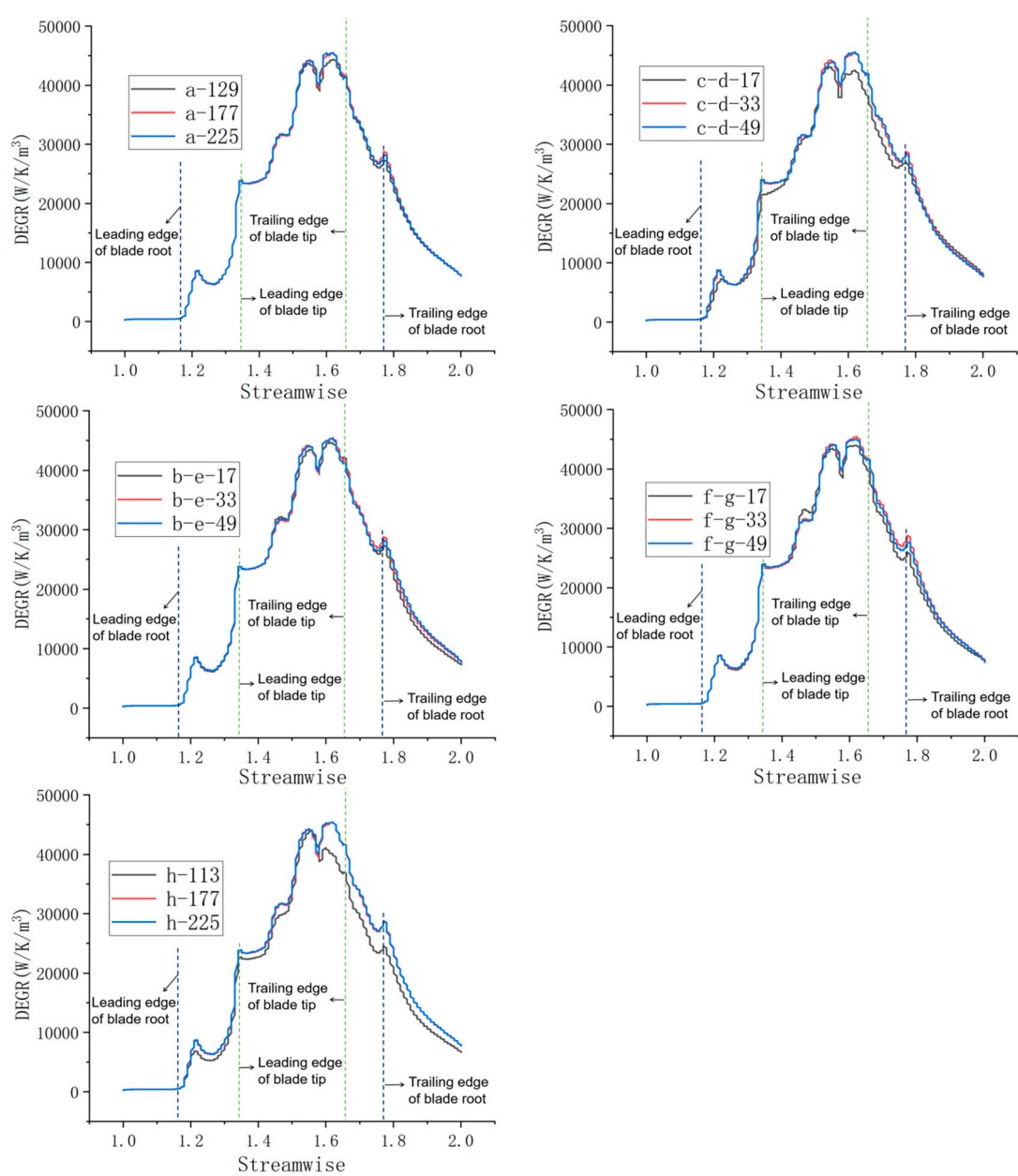

**Figure 3.** DEGR distribution along the axial direction of different grids.

### 2.2. Validation of Numerical Simulation Method

The numerical simulation was validated by comparing the numerical simulation results with experimental data. Figure 4 shows the rotor characteristic line, where the experimental data were obtained from a report published by NASA in 1991 [28]. The simulated rotor speed in this study was a design speed of 16,043 rpm, while the experimental value was 16,169 rpm. To compare with the experimental value, based on the

compressor similarity law, the speed of 16,043 rpm was corrected to 16,169 rpm. From the graph, it can be seen that the trend in the numerical simulation results is consistent with the experimental values, and the deviation falls within an acceptable range. Therefore, it can be concluded that the numerical simulation method is both feasible and effective. All subsequent calculations presented in this paper were based on the design speed.

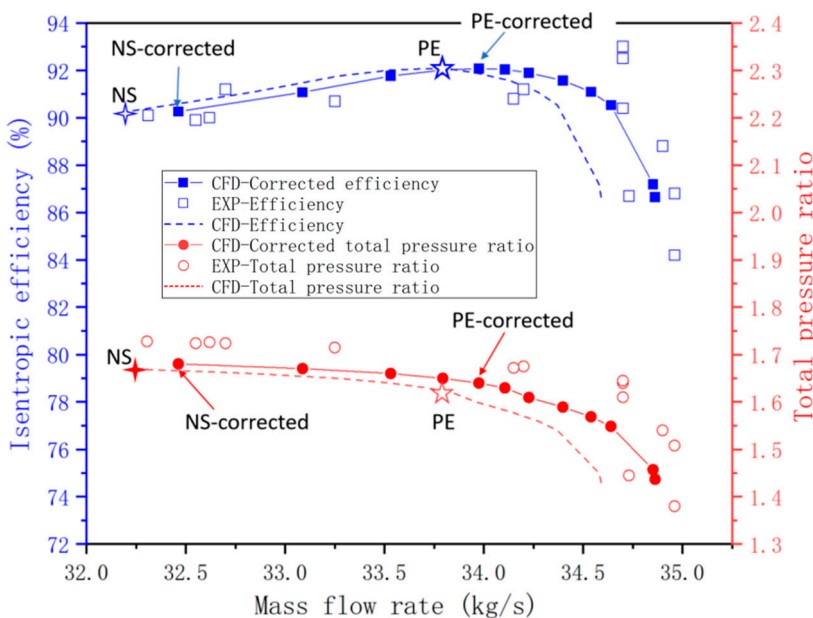

**Figure 4.** Compressor characteristics validated using experimental data.

## 3. Results and Discussions

In this section, two examples of single circumferential groove are presented on the compressor's casing, one along the blade's leading edge at the tip, the other downstream next to the first one. They are used to alter the flow structures in near tip region and provide testimonies for DEGR's connection in both efficiency and stability. This section is divided into three subsections. After introducing the two circumferential grooves, the correlation between DEGR and peak efficiency is presented and discussed first, followed by the correlation between DEGR and stability.

### 3.1. Casing Treatment Configuration and Comparison with a Solid Wall

Among many possible choices of CTs, the circumferential groove is chosen because it can reduce unsteadiness and computation time. Two such CTs with stability enhancement effects are selected, and the results are compared with those of the smooth wall (SW). In this study, the structural parameters of the CT were determined based on previous experience. The two CTs were located at streamwise positions, accounting for 0.05–0.22 and 0.22–0.39 blade chord length of the blade tip, respectively. Figure 5 shows the positions of the CTs, presented as a bi-channel configuration. The groove depth was five times greater than the blade tip clearance. In terms of meshing, the grid density at the casing remained unchanged for comparison with SW. The size of the grid nodes in the CT region was consistent with the casing surface. General connection was applied at the interface between the CT and the casing. The number of grids in a single channel of the CT was approximately 7850.

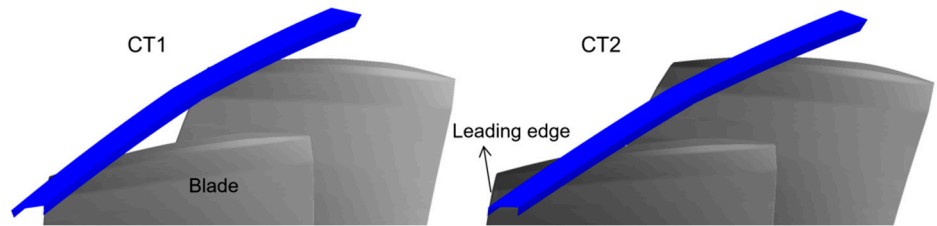

**Figure 5.** The locations of two casing treatments.

The performance characteristic curve with CTs was compared with the SW (as shown in Figure 6). These two CTs were selected because they exert different influences on the compressor flow field. As shown in Figure 6, both CTs improved the stall margin, with margin extensions of 12.02% and 8.32%, respectively. Regarding the PE condition, compared with the SW, CT1 showed almost no change, with a reduction of only 0.04%, while CT2 exhibited an increase of 0.15%. The differences in the effects of these two CTs on efficiency are beneficial for analyzing the correlation between DGER and efficiency, while differences in the effects on expansion margin are helpful for analyzing the correlation between DEGR and the capability for stability enhancement. The operating points used are labeled in Figure 6: CT1-A and CT2-A correspond to the same flow rate as SW-PE; CT1-B and CT2-B correspond to the same flow rate as SW-NS.

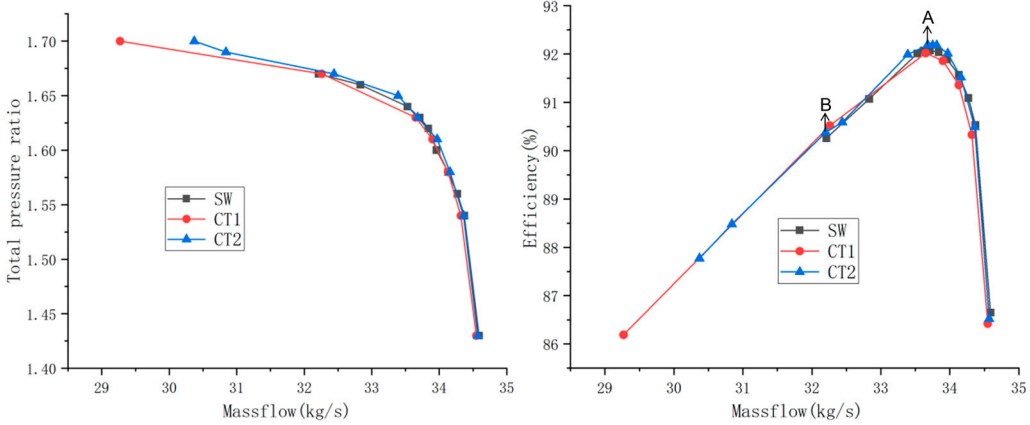

**Figure 6.** Characteristic lines of R67 with and without CTs.

*3.2. Correlation between Entropy Generation and Peak Efficiency*

In this subsection, the curves of total entropy generation versus the mass flowrate for three different casings are plotted in the same figure as the curves of isentropic efficiency, and then explained with the flow mechanism using DEGR contours. This subsection ends with a discussion about how to utilize the DEGR as a measure to compare (or even predict) the CTs' effects on efficiency.

3.2.1. Correlation of Efficiency and DEGR and its Flow Mechanism

The curves of isentropic efficiency with mass flowrate are presented in the top half of Figure 7 for three different casings: SW, CT1, and CT2. CT1 and CT2, where CT1 exhibits the best extension of stall margin and CT2 improves the efficiency the best at PE. Total entropy generation over the entire rotor section for three casings are depicted in the lower half of Figure 7, with its values on the second vertical axis on the right of the figure. It can clearly be seen that the curves of entropy generation exhibit a reverse trend against the curves of isentropic efficiency. This is not a surprise because entropy generation represents irreversibility in the flow field, which is in contrast to efficiency. More interestingly are the details in the flow fields that govern the trends.

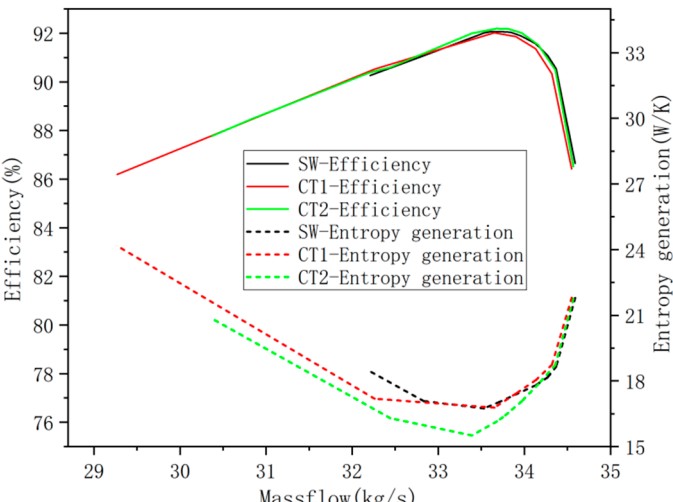

**Figure 7.** The efficiency and entropy generation distribution curves with mass flow.

The influence of CTs on the flow field is presented in Figures 8–12, among which Figures 8–10 confirm the blade tip region is the most influential and Figures 11 and 12 give the details of the flow structures presented with DEGR contours. Figure 8 illustrates the DEGR distribution on the meridional plane for SW and CTs, revealing high DEGR regions in SW-PE. The figure is obtained by averaging over the circumferential direction on the meridional plane, which cannot reflect detail differences circumferentially, but can present a comprehensive distribution of the DEGR along the span of the blade. The figure illustrates that the high DEGR regions for SW and both CTs at peak efficiency are concentrated near the blade tip. A small portion of the high DEGR region exists close to the hub of the leading edge in SW and CTs, which means that CTs do not alter the DEGR in this hub area. The impact of CTs on the DEGR is focused on the tip region. Notably, only CT2 is capable of effectively reducing the size of regions of high DEGR areas near the tip.

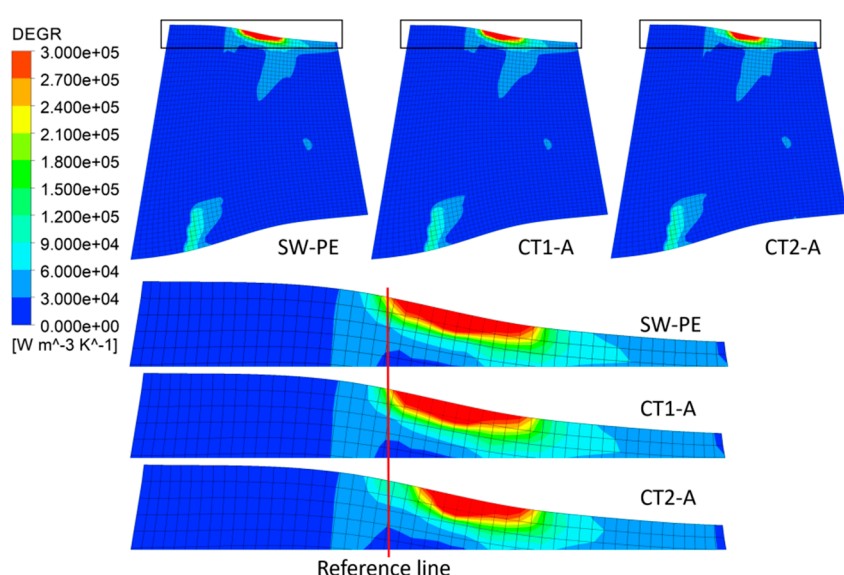

**Figure 8.** DEGR distribution on the meridian plane and magnified image of the tip region.

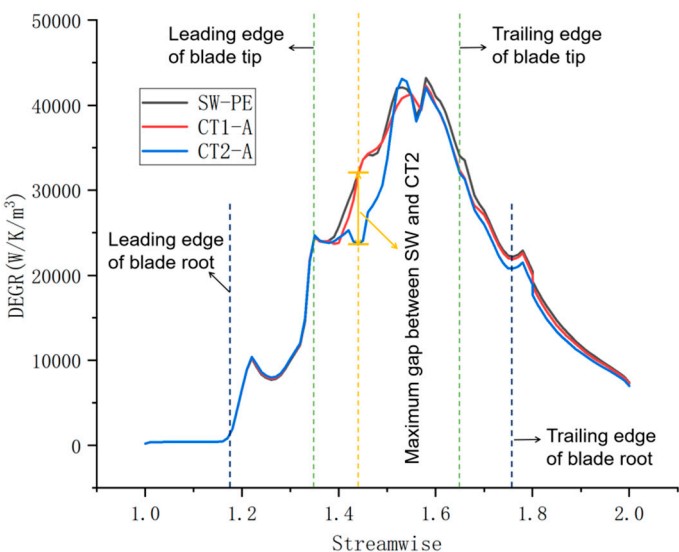

**Figure 9.** Spanwise-averaged DEGR distribution along the axial direction.

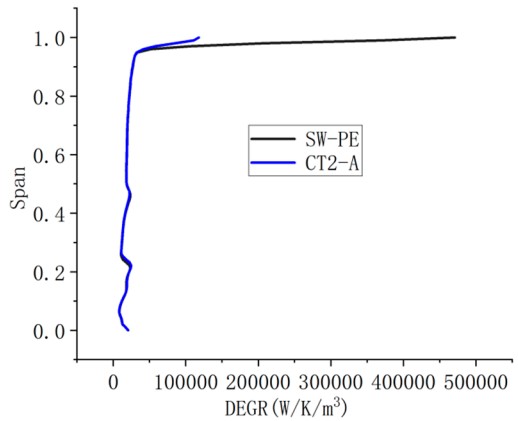

**Figure 10.** DEGR distribution on 1.44 streamwise of SW-PE and CT2-A.

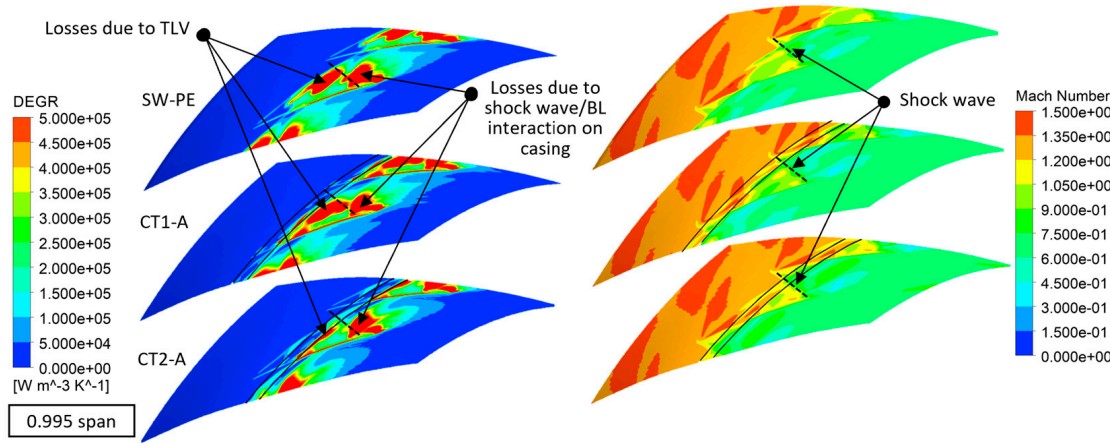

**Figure 11.** DEGR distribution and the relative Mach number.

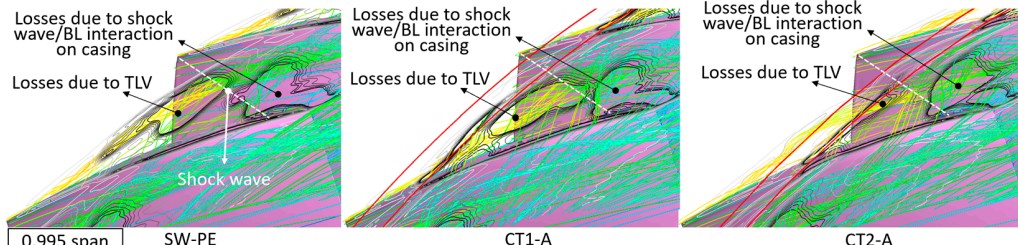

**Figure 12.** DEGR distribution and three-dimensional streamline.

To more precisely determine the specific zone of the highest DEGR, a two-step process was employed. Firstly, the DEGR distribution along the streamwise direction between SW and CTs was plotted (Figure 9). No significant difference was observed between SW and CT1, but there was a gap between SW and CT2. This result is consistent with the efficiency data. Then, the spanwise distribution of the averaged DEGR at a streamwise location at 1.44 (as marked in Figure 9) was plotted in Figure 10, indicating that the discrepancy between SW and CT2 lay in the top 5% of the blade, within the range of 95–100% span. The curve for CT1 is the almost same as SW, and thus, was not plotted.

The subsequent section provides a detailed analysis of the flow structure changes in the tip region of CTs, while Section 3.2.2 focuses on measuring the efficiency enhancement ability of CTs within the 0.95–1 span region.

The tip region of a compressor represents the most complex flow area, where losses arise from various phenomena such as tip leakage vortex (TLV), shock wave/boundary layer (BL) interaction on casing and blade, among others. Previous research conducted by Ma et al. [22] investigated the distribution of DEGR corresponding to these flow structures. Building on their findings, this study delved deeper into the flow mechanisms that are impacted by changes in DEGR distribution modified by CTs.

Figure 11 depicts the DEGR distribution and relative Mach number, with dashed lines indicating shock wave positions determined based on the distribution of relative Mach numbers, while black lines represent groove positions. The chosen cross-section for contouring the DEGR distribution is located at 0.995 h, which is situated in the middle of the blade tip gap where the TLV and shock wave/BL interaction on the casing are the primary flow structures influencing the DEGR on this surface. Selecting this particular cross-section enabled the better differentiation of losses caused by TLV and the shock wave/BL interaction [22]. Figure 11 indicates that CT2 significantly reduces the area of high DEGR attributed to TLV in comparison with SW. However, both CTs had no significant impact on the high DEGR caused by shock wave/BL interactions.

To further investigate the flow mechanism responsible for differences in DEGR changes between these two CTs, Figure 12 presents a transparent rendering of the DEGR distribution over three-dimensional streamlines near the tip region. The DEGR distribution in this figure is identical to that shown in Figure 11, with the only difference being the method of representation. By comparing the two figures, it can be observed that areas with high DEGRs are represented by the inner regions of the black outlines. The groove position is represented by a red line. The pink region represents the blades and the pressure surface. The left side denotes the inlet direction.

From Figures 11 and 12, it can be observed that the high DEGR region is formed by the interaction between TLV and leakage flows in SW. The groove position of CT2 is located just above the region where TLV generates a high DEGR, weakening the leakage flow in this area, and thus reducing the DEGR. In contrast, the groove position of CT1 is located upstream of the high DEGR region caused by TLV, and therefore has little effect on the highest DEGR.

### 3.2.2. The Metric of the CTs Efficiency Change by DEGR

At this point, the relationship between the changes in DEGR, the tip flow structures and the efficiency of the compressor has been thoroughly explored. However, a more intuitive metric for comparing the impact of different CTs on efficiency is still lacking. It has been shown that the impact of CTs on the DEGR distribution in the compressor is concentrated in the blade tip region, and the DEGR within the near-blade tip region can be a metric to compare the irreversibility between CTs. Here, the range above the 0.95 span is chosen for averaging and comparing the DEGR. The results of such averaging are presented in Figure 13 for SW-PE and CTs-A. The vertical axis DEGR represents the average values along the circumferential direction in the 0.95–1 span region. Figure 13a shows the local DEGR, while Figure 13b represents the cumulative DEGR along the streamwise direction. From the local DEGR values, it can be observed that CT2, which improves efficiency, reduces the DEGR throughout the entire chord length of the blade, with the most significant improvement occurring in the region after the location of the groove. From the cumulative DEGR values, it can be seen that CT2 demonstrates the smallest cumulative DEGR reduction, indicating a superior ability to reduce irreversible losses, and correspondingly, improve efficiency. In comparison, CT1 only reduces DEGR in certain regions of the blade's chord length, and also reduces the cumulative DEGR, indicating an improved flow field. However, these changes are not sufficient to significantly alter efficiency.

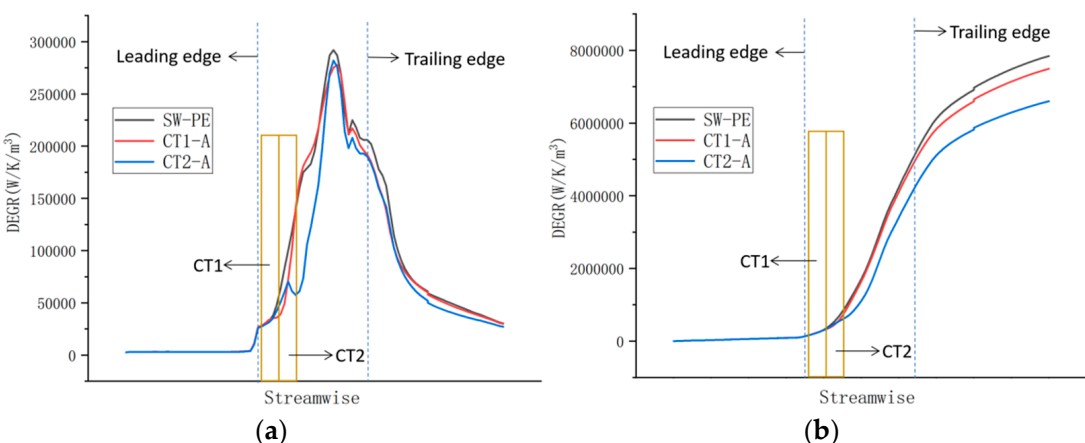

**Figure 13.** DEGR distribution above the 0.95 span along with streamwise. (**a**) Local quantity; (**b**) cumulant quantity.

Henceforward, it can be concluded that DEGR can be utilized as a measure to quantify and compare the ability of different CTs to enhance peak efficiency. The method is as follows: compare SW-PE with the corresponding CT case at the same mass flow along the streamwise direction, and only when there is a significant reduction in DEGR distribution, it is possible to achieve an improvement in efficiency simultaneously.

### 3.3. Correlation between the DEGR and the Stability

Although it is quite straightforward to relate entropy generation to flow losses, and thus, efficiency, it is not natural to measure flow instability with the DEGR. It needs to recall theories about stall onset over the last twenty years. It is argued that in tip-sensitive rotors, such as R67 (as shown in Figure 10), stall is triggered by the tip leakage flow (TLF) that initiates the reversed flow against the main flow (MF). An MF/LF interface exists, and stall cells emerge once this interface moves forward and spits out of the blade's leading edge. The intuitive method to visualize this interface is to plot the lines of wall shear on the surface of the casing, which would exhibit the incoming flow and the reverse flow and therefore indicate the line that separates the two flows. In this section, we argue that DEGR

contours can also be used to identify the MF/LF interface, because the entropy generation should be very small in MF and then jump sharply in the flow field of TLF. The more unstable the rotor (with or without casing treatments), the closer the interface to the blade leading edge.

3.3.1. Connection between DEGR and the MF/LF Interface at the near Stall

The fundamental nature of the onset of the compressor stall is characterized by the gradual displacement of the MF/LF interface towards the leading edge of blades. Previous research [29] has established that a decrease in mass flow causes the wall shear boundary on the casing surface to migrate towards upstream blade edges, ultimately resulting in compressor flow instability. Essentially, the migration of the wall shear boundary on the casing surface serves as an indicator for distinguishing between mainstream flow and tip leakage flow. This subsection investigates whether DEGR can function as a marker for the MF/LF interface, with the wall shear boundary serving as the connection between the MF/LF interface and DEGR.

Figure 14 illustrates the distribution of the DEGR and the lines of wall shear on the casing surface. The boundary of wall shear, which separates MF and TLF, is clearly distinguishable from the leading edge to the pressure surface of the upstream blade. It moves forward from PE to NS operating conditions.

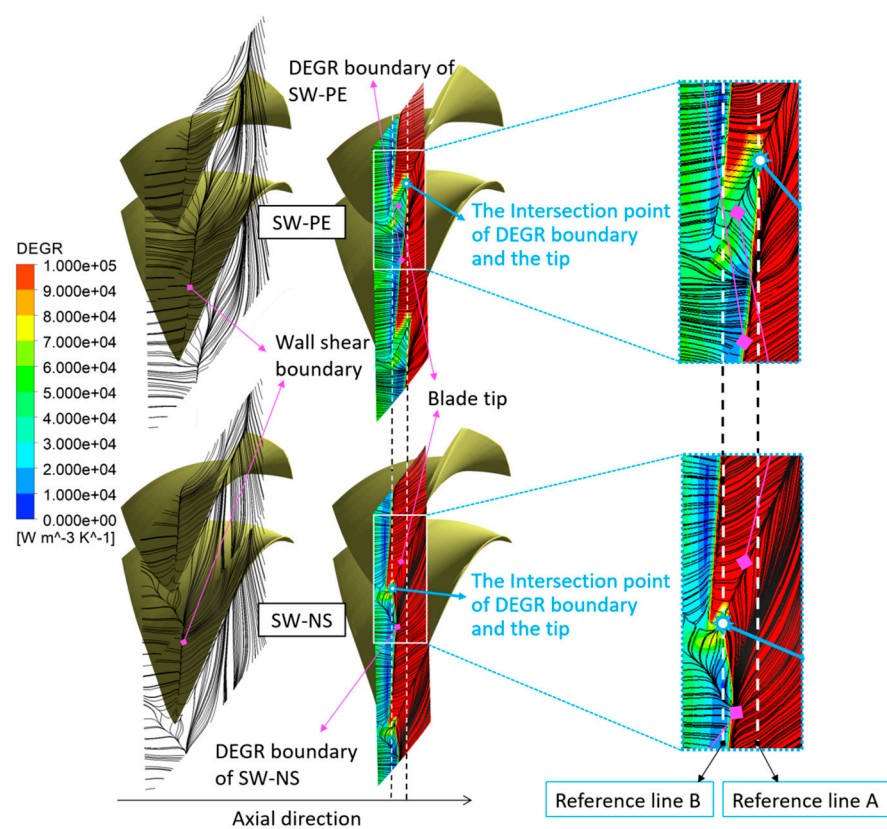

**Figure 14.** The wall shear line and DEGR distribution on the shroud.

In Figure 14, the map of DEGR contours is plotted, which overlaps onto the wall shear lines. The discontinuity observed in DEGR was found to be consistent with the wall shear boundary, both in PE or NS operating conditions, which demonstrates that the boundary of discontinuity of DEGR can also serve as an indicator for the MF/LF interface. To quantitively compare the locations of the interface, a magnified view is placed on the right side, on which two blue points that represent the intersection of DEGR boundary and the blade tip are marked. The right point corresponds to the DEGR boundary in the PE operating condition, while the left point corresponds to the NS operating condition. The

white dotted lines indicate circumferential lines passing through these intersection points and are marked with reference lines A and B. These lines serve to illustrate the forward displacement of the DEGR boundary from PE to NS operating conditions.

Notably, the selection of the maximum value for the DEGR scale in Figure 14 is not arbitrary. It was necessary to capture the first sharp increase in the DEGR curve (Figure 16). For this study, we chose a value of 100,000 (W/K/m3) for the DEGR contours, and consistently used it when comparing with other entropy generation flow fields with different CTs in subsequent subsections. This choice is justified in next subsection.

The DEGR boundary offers an advantage in providing information across various regions compared with the wall shear line, which is limited to the casing surface. Since stall can emerge within a specific radial range, it is crucial to comprehensively investigate the characteristics of the DEGR boundary in other span sections as well. Figure 15 illustrates the distribution of the DEGR on different span sections using a graph scale identical to that in Figure 14. Figure 15 exhibits distinct DEGR boundaries at 0.99, 0.98, and 0.97 spans. The DEGR boundary located at a height of 0.98 span was used as an indicator for the MF/LF interface under full-speed conditions in this particular compressor model. The same span was employed to perform comparisons with other entropy generation flow fields featuring different CTs in subsequent sections.

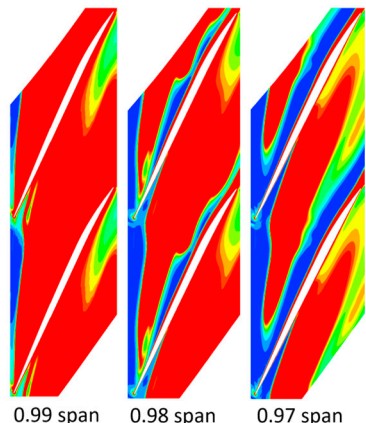

0.99 span    0.98 span    0.97 span

**Figure 15.** DGR boundaries of three spans for SW-NS.

### 3.3.2. The Metric of the CTs Stability Enhancement by DEGR

In the previous subsection, it was argued that the DEGR boundary is effective for measuring the MF/LF interface. The contours at a radial height of 0.98 span are chosen for comparison purposes. This subsection utilizes the DEGR boundary to provide a qualitative comparison of CTs' impact on stability enhancement. To facilitate the measurement of stability enhancement capabilities among different CTs, it is imperative to quantify this DEGR boundary line as a single value, rather than an inclining line on the contour map. The proposed approach involves averaging the DEGR within the radial influence range of the CTs and then circumferentially, resulting in precise DEGR values at each position along the streamwise direction.

Two figures have been generated to analyze the impact of different CTs on stability enhancement. As these two figures are closely related, they have been combined into a single figure for ease of explanation. Figure 16 is divided into two parts: the upper part represents the distribution of radially and circumferentially averaged DEGR along the streamwise direction of both SW-NS and CTs-B cases above the 0.95 span and up to the shroud (1.0 span). The blue dashed lines indicate the positions of the leading and trailing edges of the blade tip, while the yellow rectangles represent the streamwise position ranges for two CT cases. The pink dashed line represents the boundary value of the DEGR (100,000 W/K/m$^3$). Given that CT1 has a more anterior groove position, it may be inferred that its effect in shifting the MF/LF interface is greater. The upper part of Figure 16 confirms

this, as the three intersection points between the DEGR curve and the pink dashed line near the leading edge indicate the average MF/LF interface for SW (black), CT1 (red), and CT2 (blue), respectively. The corresponding streamwise values denote their respective positions. The upper part of the figure indicates that both the red and blue circles are positioned behind the SW point, with CT1 (the red circle) having the highest streamwise value and being located at the most posterior position, followed by CT2 (the blue circle). This outcome is consistent with the expected judgment and also correlates with stall margin results of the CTs. The lower part of Figure 16 displays DEGR distributions on the 0.98 span section for these three cases (SW, CT1, and CT2). The CT1-B groove is positioned just above the original DEGR boundary of SW-B, resulting in a low DEGR region below it that effectively inhibits forward movement of the DEGR boundary. In CT2-B, the groove is located behind the original DEGR boundary of SW-B and improves a small area on the pressure side of the blade to form a low DEGR zone, slightly shifting back the stall boundary line in this region.

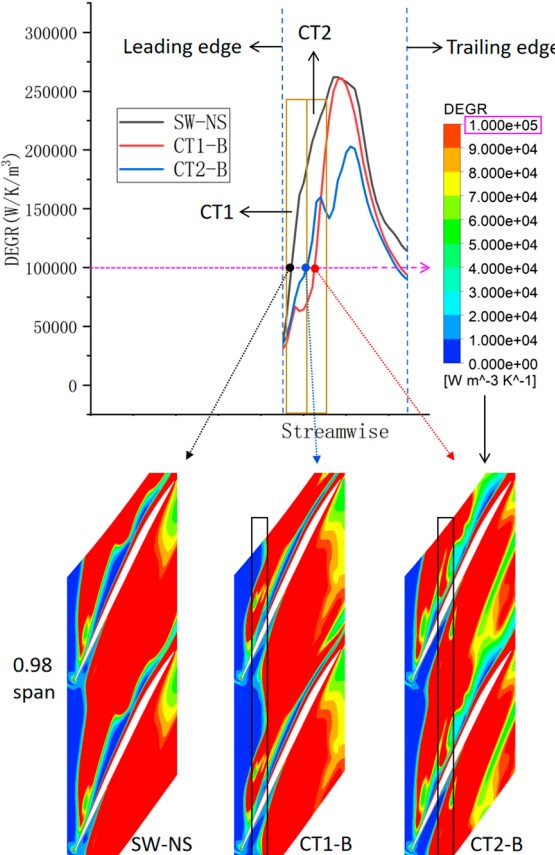

**Figure 16.** The upper half: DEGR distribution above the 0.95 span along with the axial direction. The bottom half: DEGR distribution at the 0.98 span.

We have developed a methodology for assessing the stability enhancement of CTs. In order to optimize their design, it is crucial to understand the flow mechanism between DEGR and stability enhancement. The following subsection explores this relationship in detail.

### 3.3.3. Flow Mechanism between the DEGR and Stability Enhancement

The flow structures in the vicinity of the blade tip during the transition from PE to NS operating conditions are the focus of this subsection. It has been demonstrated in many studies [3,6–8,30] that as adverse pressure gradients increase, the angle between the TLV and axial direction also increases, causing TLV to propagate upstream. Simultaneously, the shock waves' location on the blade's suction face shifts upstream, resulting in a deteriorated

flow field, and subsequently, increased irreversible losses. In light of these two common flow structures, what role do stability-enhancing CTs play? Figures 17 and 18 provide answers to this inquiry.

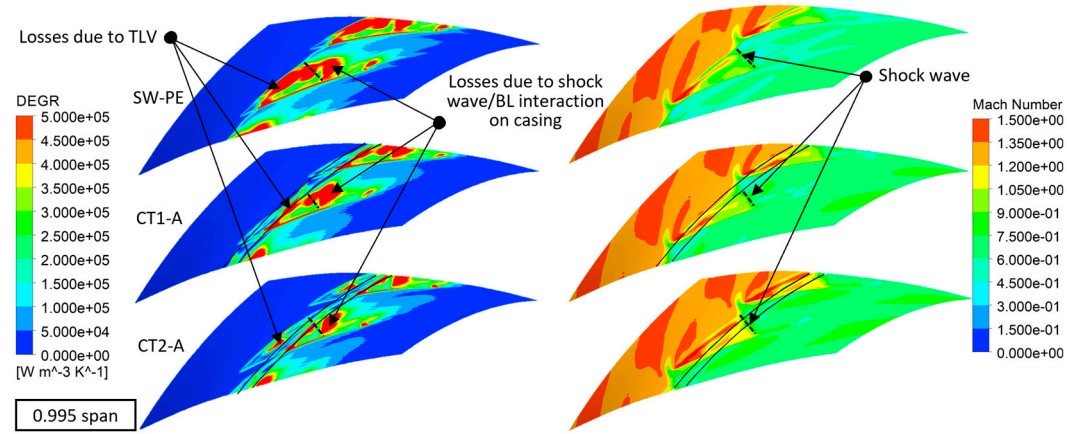

**Figure 17.** DEGR distribution and relative Mach number on the 0.995 span.

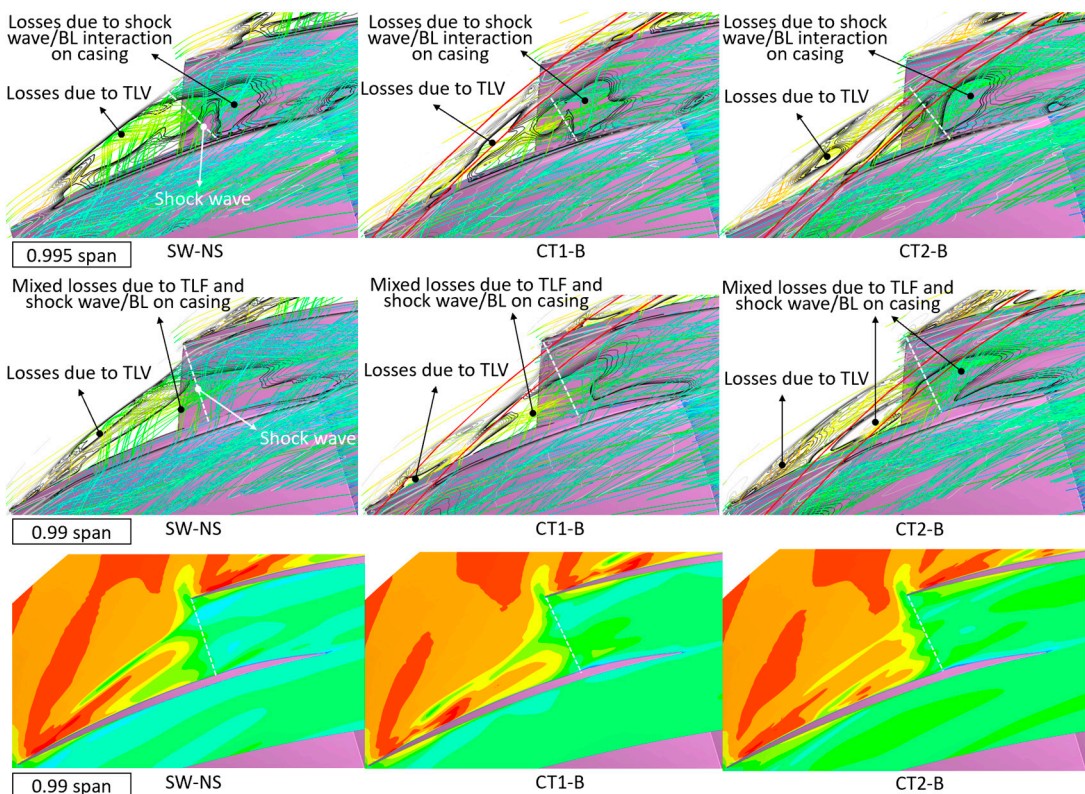

**Figure 18.** DEGR distribution and three-dimensional streamline with the relative Mach number contour.

In Figure 17, the cross-sectional observation of flow structures is selected at the 0.995 span to facilitate comparison with Figure 11. The figure illustrates the DEGR and relative Mach number distributions on the 0.995 span section for SW and CTs, with black dashed lines indicating shock wave positions derived from the relative Mach number distribution; groove positions are also marked. In Figure 17, it is evident that the high DEGR region induced by TLV expands on the 0.995 span section of SW-NS in comparison to SW-PE (as shown in Figure 11). Both the TLV and shock wave/boundary layer (BL) interaction cause upstream movement of their respective high DEGR regions. CT2 exhibits a superior

ability to mitigate the DEGR on this specific height section. This is due to the fact that CT2's groove position covers a portion of both the TLV and shock wave locations, while CT1 primarily mitigates the high DEGR caused by TLV and has a negligible impact on the high DEGR region caused by shock wave/BL interactions. To provide a more intuitive depiction of the relationship between TLV and DEGR, Figure 18 illustrates the distribution of DEGR on the 0.995 span and 0.99 span, overlapping with the three-dimensional streamlines with the relative Mach number contour on the 0.99 span for SW and CTs. The label is same as in Figure 17. It can be seen that the groove of CT1 effectively suppresses the expansion and the growth of TLV, whereas the groove position of CT2 is situated further back and only partially inhibits TLV. It has little impact on the high DEGR caused by the front half of the TLV. Moreover, the groove of CT2 is positioned above specific shock wave positions and primarily functions to reduce shock wave/BL interactions.

### 4. Conclusions

In conclusion, the DEGR can serve as a unified measure to evaluate both efficiency and stability. Several points are made as follows:

(1) The total entropy generation shows a reversed trend to isentropic efficiency as mass flow varies. In the PE operating condition, high DEGR regions are primarily concentrated near the blade tip. The influence of CTs on DEGR is also focused around the blade tip. CTs can enhance efficiency by suppressing the highest DEGR generated by complex flows around the blade tip.

(2) The DEGR boundary aligns completely with the wall shear boundary at the near stall, which means that the DEGR boundary can represent the MF/LF interface. At the near stall, high DEGR regions are mainly concentrated near the blade tip, and the influence of CTs on DEGR is also focused around the blade tip. CTs can narrow down the range of high DEGR and push the DEGR boundary downstream. The flow mechanism shows that the CTs inhibit the TLV turning toward to the leading edge. In other words, CTs suppress the interface of MF/LF moving upstream, thereby delaying the onset of stall.

(3) The method of how to utilize the DEGR to measure the efficiency and stability enhancement of CTs are proposed. Volumes within the 0.95–1 span were chosen for averaging the DEGR. The cumulative distribution of the DEGR along the axial direction provides a measure for efficiency improvement ability of CTs. The location of the DEGR along the axial direction provides a measure for the stability enhancement ability of CTs.

**Author Contributions:** J.Y.M. performed all the numerical computations and wrote Sections 2 and 3. F.L. proposed the research ideas and wrote the remaining sections. All authors have read and agreed to the published version of the manuscript.

**Funding:** This project was supported by the Department of Science and Technology of Fujian Province (grant number 2020H6015), the Department of Science and Technology of Fujian Province (grant number 2022I0018), the Jimei University research start-up fund (Jingyuan Ma, 2021.3-2024.3), and the Jimei University research start-up fund (Feng Lin, 2020.12-2025.12).

**Data Availability Statement:** The data that support the findings of this study are available from the corresponding author upon reasonable request.

**Conflicts of Interest:** The authors declare that they have no known competing financial interests or personal relationships that could have appeared to influence the work reported in this paper.

### Nomenclature

| | |
|---|---|
| $\alpha$ | Thermal diffusivity, (m$^2$/s) |
| $\alpha_t$ | Thermal diffusivity of the fluctuating temperature, (m$^2$/s) |
| $\theta$ | Dimensionless temperature, K |
| $\lambda$ | Thermal conductivity, J s$^{-1}$ m$^{-1}$ K$^{-1}$ |

| | |
|---|---|
| $\mu$ | Dynamic viscosity, kg m$^{-1}$ s$^{-1}$ |
| $\Phi_\theta$ | Entropy production term, (WK/m$^3$) |
| $\omega$ | Characteristic frequency, MHz |
| $n$ | Rotation speed, rpm |
| $p$ | Local pressure, N m$^{-2}$ |
| $\dot{S}_{irr,D\prime}^{\prime\prime\prime}$ | Entropy production rate by turbulent dissipation, (W/(m$^3$ K)) |
| $\dot{S}_{irr,D}^{\prime\prime\prime}$ | Entropy production rate by viscous dissipation, (W/(m$^3$ K)) |
| $\dot{S}_{PRO,C\prime}$ | Entropy production rate by heat transfer with gradients of the fluctuating temperature, (W/(m$^3$ K)) |
| $\dot{S}_{PRO,C}$ | Entropy production rate by heat transfer with mean temperature gradients, (W/(m$^3$ K)) |
| $T$ | Bulk temperature, K |
| u′ v′ w′ | Local fluctuating velocity component, m s$^{-1}$ |
| u  v  w | Local average velocity component, m s$^{-1}$ |
| x y z | Coordinate vector component, m |

**Abbreviations**

| | |
|---|---|
| CT | Casing treatment |
| CT-A | Casing treatment's A operating condition |
| CT-B | Casing treatment's B operating condition |
| TLV | Tip leakage vortex |
| DEGR | Differential entropy generation rate |
| SW | Solid wall |
| EXP | Experimental data |
| NS | Near stall |
| NS-corrected | Corrected near stall |
| PE | Peak efficiency |
| PE-corrected | Corrected peak efficiency |
| R67 | NASA Rotor 67 |
| BL | Boundary leakage |
| MF/LF | Main flow/leakage flow |
| Subscripts | |
| gen | Generation rate |
| $irr, D\prime$ | Turbulent dissipation |
| $irr, D$ | Viscous dissipation |
| $PRO, C\prime$ | Heat transfer with gradients of the fluctuating temperature |
| $PRO, C$ | Heat transfer with mean temperature gradients |
| $rev$ | Reversible |
| $t$ | Relative total |

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
