# Peer review of "The Differential Entropy Generation Rate as a Unified Measure for Both the Stability and Efficiency of an Axial Compressor"

_machines, doi:10.3390/machines11080815_

Round 1
Reviewer 1 Report
Authors have presented numerical analysis for stability and efficiency of an axial flow compressor. Their analysis results are closer to the experiments with least margin of difference.
Introduction is well explained although few more references could be added. In the section 2, authors can improvise their model description with adding some forumlations which are used (in mathematical). Results, Discussions and conclusions are very well written.
The paper can be accepted after making minor changes in sections 1 and 2.
Reviewer 2 Report
The paper needs minor revision.
1. What is entropy and entropy generation rate physically? Write a few lines on these in the introduction.
2. Explain the value from Table 1 .
3. What are the convergence criteria of numerical technique.
4. The authors should add a table of symbols in the revised manuscript.
5. Cite all of these numerical-based studies:
*Locomotion of an efficient biomechanical sperm through viscoelastic medium
*Dynamical interaction effects on soft-bodied organisms in a multi-sinusoidal passage
*Controlling kinetics of self-propelled rod-like swimmers near multi sinusoidal substrate
*Biomechanics of bacterial gliding motion with Oldroyd-4 constant slime
*Enhancing motility of micro-swimmers via electric and dynamical interaction effects
*A numerical framework for modeling the dynamics of micro-organism movement on Carreau-Yasuda layer
*An exact solution for directional cell movement over Jeffrey slime layer with surface roughness effects
Minor revision is needed.
Reviewer 3 Report
Review of Article 1: "The differential entropy generation rate as a unified measure for both stability and efficiency of an axial compressor."
The article presents a study focused on highly loaded aviation axial compressors, where stability and efficiency are crucial performance indicators. The authors propose the use of the differential entropy generation rate as a unified measure to simultaneously consider both stability enhancement and efficiency improvement during casing treatment design. They conduct experiments on NASA Rotor 67, applying two single circumferential grooves at different axial positions as test cases to observe how the entropy generation rates in the flow field change concerning peak efficiency and stall margin.
The use of the differential entropy generation rate as a unified measure is an interesting approach to address the trade-off between stability and efficiency in axial compressors. The idea of casing treatment and its potential to improve stability at the expense of peak efficiency is a relevant problem in the field, and the authors attempt to tackle it with a quantifiable metric.
Strengths:
Novel Approach: The use of differential entropy generation rate as a unified measure is a fresh and innovative concept that could potentially have significant implications for axial compressor design and performance optimization.
Appropriate Test Cases: The selection of NASA Rotor 67 and the application of circumferential grooves at different axial positions as test cases seem appropriate for examining the proposed unified measure's effectiveness.
Comprehensive Analysis: The article covers a wide range of aspects, including the correlation between differential entropy generation rate and peak efficiency, flow mechanism analysis, impact measurement of different casing treatments, and exploration of differential entropy generation rate in near stall conditions.
Questions for the Authors:
Validation of Unified Measure: How extensively was the proposed differential entropy generation rate metric validated against other existing measures of stability and efficiency in axial compressors? Can you provide more insights into how the new measure outperforms or complements traditional indicators?
Practical Applicability: In real-world engineering scenarios, implementing casing treatments can be complex and costly. Have you considered the practical aspects of applying the proposed unified measure in actual axial compressor designs? Are there any potential challenges or limitations in using this metric for industry applications?
Overall, the article presents a promising and innovative approach to addressing the stability and efficiency trade-off in axial compressors. However, it would be beneficial to further validate the proposed unified measure against existing metrics and discuss its practical applicability to enhance its credibility and potential real-world impact.
